# Lateral Alveolar Ridge Augmentation with Autologous Dentin of Periodontally Compromised Teeth: A Retrospective Study

**DOI:** 10.3390/ijerph19084560

**Published:** 2022-04-10

**Authors:** Michael Korsch, Marco Peichl, Andreas Bartols

**Affiliations:** 1Dental Academy for Continuing Professional Development, Lorenzstrasse 7, 76135 Karlsruhe, Germany; marco_peichl@za-karlsruhe.de (M.P.); andreas_bartols@za-karlsruhe.de (A.B.); 2Clinic of Operative Dentistry, Periodontology and Preventive Dentistry, University Hospital, Saarland University, Building 73, 66421 Homburg, Germany; 3Private Practice, Center for Implantology and Oral Surgery, Berliner Str. 41, 69120 Heidelberg, Germany; 4School for Dental Medicine, Christian-Albrechts-University Kiel, Clinic for Conservative Dentistry and Periodontology, 24105 Kiel, Germany

**Keywords:** dental implant, bone graft, autologous, dentin graft, buccal bone, periodontitis

## Abstract

Tooth shell technique (TST) using autologous dentine is possible with lateral ridge augmentation while avoiding a donor region. This study aimed to clarify whether the use of periodontally compromised teeth (PCT) leads to similar results compared to non-periodontally compromised teeth (NPCT). In this retrospective study, the dentin matrix of 41 patients (PCT: *n* = 19 with 29 implants; NPCT: *n* = 22, with 29 implants) was used for TST. All cases were re-examined. Outcome parameters were biological complications, horizontal hard tissue loss, osseointegration, and the integrity of the buccal lamella. Only in one case in the PCT group, a graft was lost. In three cases, minor complications were identified, including two cases of wound dehiscence and one case of inflammation with suppuration (PCT: *n* = 1, NPCT: *n* = 3). All implants, except the one with the severe complication, were osseointegrated and the integrity of the buccal bone lamella was preserved. Mean difference of the resorption of the crestal width and the buccal lamella did not differ statistically between the two groups. TST using PCT showed results comparable to those of NPCT in terms of complications and graft resorption. Processed dentin matrix from PCT can be used and applied with predictable results for bone grafting, utilizing TST.

## 1. Introduction

In the treatment of bone defects in the alveolar ridge that do not allow for primary implant placement, bone augmentation must be performed first. In recent years, techniques using autologous dentin for bone augmentation have been increasingly described for this purpose. The method has been reported to be successful in several publications in humans and can be deemed to be principally established [1,2,3,4]. Since dentoalveolar surgery is always performed under the principle of preventing infection and under aseptic conditions as possible, dentin from uninfected teeth was used in these studies. Mainly caries- and filling-free wisdom teeth or impacted teeth were applied [1,2,3,4] to reduce the risk of infection from the graft material to a minimum. However, this restricts the technique to a limited number of patients who have suitable teeth, though this does not reflect the clinical reality. In many patients, a much more common scenario is that a tooth cannot be retained for various reasons, yet it is to be implanted in the same position afterwards. In such cases, there is not always another supernumerary, non-compromised tooth available that can serve as grafting material. The tooth to be removed is often the only possibility to obtain autologous dentin for augmentation. However, it is to be expected that teeth that cannot be preserved will have some type of infectious disease [5,6].

In two animal studies, it was shown that the use of dentin from endodontically-treated or periodontally-compromised tooth roots is in principle suitable as an augmentation material for alveolar ridge defects. However, a somewhat higher rate of wound dehiscences was found in compromised teeth compared to cases with uncompromised roots or autogenous bone blocks. Although the dehiscent sites showed no clinical signs of infection, all the grafts were lost and subsequent implant placement was not possible. However, the authors did not report whether the dehiscence rates were statistically significantly different between the experimental groups [7,8]. In a recent retrospective study of our research group, the augmentation of alveolar bone defects in humans with dentin from endodontically treated teeth was compared with that of uncompromised teeth. No relevant differences were found with regard to wound healing disorders. In general, the complication rate was very low. No graft was lost and implant placement was possible in all treatment cases [9].

Even though these results are encouraging, there is a knowledge gap regarding the use of periodontally compromised teeth. As a rule, periodontally damaged teeth have an infection problem, similar to endodontically treated teeth. Thus, periodontally compromised teeth often have calculus on the root surface that harbours bacterial infection [10]. Since artificially induced periodontitis was investigated in the above-mentioned animal study [8], which might be different from an established chronic periodontal infection in humans, with bacteria possibly penetrating beyond the cementum layer and into the depths of the dentinal tubules [11], the question arises as to whether the preparation and disinfection of the dentin provide sufficient microbial reduction. To ensure the success of the augmentation as much as possible, wound infection should be prevented. Infections can jeopardise wound closure and lead to dehiscence or, in the worst case, result in the loss of the augmentation [12]. Given this background, infected dentin, therefore, appears unsuitable for augmentation at first glance.

The present study is a retrospective proof-of-concept study in which bone augmentations with autologous dentin were performed using the tooth-shell technique (TST) [1,2] analogous to the shell technique according to Khoury [13]. Augmentations with dentin from periodontally compromised teeth and with dentin from periodontally healthy teeth were compared. The aim was to investigate whether dentin from periodontally compromised teeth caused more complications and failures in bone augmentation than dentin from periodontally healthy teeth.

## 2. Materials and Methods

### 2.1. Study Design

This retrospective study included all patient cases that underwent bone augmentation with either dentin from periodontally compromised teeth or periodontally healthy teeth between January 2019 and March 2020. For this purpose, the electronic patient records of the Outpatient Clinic of the Dental Academy for Continuing Professional Development (Karlsruhe, Germany) and the Center for Implantology and Oral Surgery (Heidelberg, Germany) were screened to identify appropriate cases. Patients were included if a follow-up over 5 months up to the completion of fixed prosthetic restoration on the implants inserted into the augmented bone was available.

Two study groups were formed: Augmentation with periodontally compromised teeth (PCT) and augmentation with non-periodontally compromised teeth (NPCT).

The study was conducted following the Declaration of Helsinki and the Professional Code of Medical Conduct of the local Medical Association. The Institutional Medical Review Board of Baden-Württemberg approved the study (ID: F-2020-068-z). The study was reported following the STROBE statement (EQUATOR guidelines).

### 2.2. Participants

All included patients had either a non-preservable tooth (a severely periodontally compromised tooth) in the prospective implant site or a tooth not worthy of preservation (e.g., a wisdom tooth) elsewhere, which was suitable for bone augmentation. In all cases, the width of the alveolar ridge was measured with a preoperative cone-beam computed tomography (CBCT). For this study, all cases with a bone defect of at least 4 mm were included, so that a hard tissue gain of 4 mm or more in width had to be achieved by augmentation. With a planned implant diameter of 4.2 mm, a total alveolar ridge width of 7.2 mm had to be obtained so that the implants were surrounded by at least 1.5 mm of hard tissue circularly.

Included patients had to be >18 years old, in need of an alveolar crest augmentation of a lateral bony defect suitable to be treated with the tooth-shell technique, had to have a lateral alveolar crest defect of at least 4 mm in the region of implant placement before augmentation with a fixed implant-retained restoration intended in an edentulous region with a maximum of three missing teeth, no untreated or residual periodontitis, no uncontrolled diabetes mellitus with HbA1c > 7%, no malignant neoplasms, no history of therapy with bisphosphonates or other antiresorptive medication (e.g., RANKL inhibitors), no history of radiotherapy in the region of head and neck, and no immunosuppression or immunosuppressant therapy.

All operations were performed by one single oral surgeon (Michael Korsch).

### 2.3. Clinical Procedure of the Tooth Processing

After extraction, the tooth scheduled for grafting was thoroughly cleaned mechanically by removing any debris/calculus and the periodontal ligament with a bur under water cooling (Figure 1a–d). A thin shell of root dentin (~1–1.5 mm thick) was obtained under water cooling with a rotating diamond wheel (Frios MicroSaw, Dentsply Sirona Implants, Mannheim, Germany). The residual dentin was crushed into particles of 300–1200 µm using a sterile disposable grinder (Smart Dentin Grinder, Kometa Bio, Creskill, NJ, USA) (Figure 1c,d). The dentin shell and the particulated dentin were placed in a sterile, closed dish containing a solution of sodium hydroxide (0.5 N, 4 mL) and ethanol (20 vol.%, 1 mL) (Dentin Cleanser, Kometa Bio, Creskill, NJ, USA) for 10 min for chemical cleaning, degreasing, and disinfection. Following the exposure time, the excess was removed with sterile gauze and the compound was washed for 3 min in phosphate-buffered physiological saline (Dulbecco’s Phosphate Buffered Saline, Kometa Bio, Creskill, NJ, USA) by manual shaking. To expose the collagen fiber network and release osteoinductive growth factors, partial demineralisation of the dentin was carried out by placing the material in a 10% EDTA solution (EDTA solution, Kometa Bio, Creskill, NJ, USA) for 3 min. The total processing time was 16 min. Again, the material was washed with buffered saline solution. Finally, it was either used immediately for augmentation or dried at moderate temperature (below 38 °C) on a hot plate and stored in a sterile container at −18 °C until transplantation. In the second case, the transplant material was defrosted at the time of grafting with the same hot plate as previously (below 38 °C) and lightly moistened with saline solution afterward.

### 2.4. Surgical Procedure of the Tooth Shell Technique (TST)

As part of perioperative antibiosis (1 day preoperatively and 2 days postoperatively), 750 mg of amoxicillin three times a day was administered. If penicillin intolerance was present, 300 mg of clindamycin was used as a substitute. For all operations, Articaine with Epinephrine 1:100,000 (Citocartin Sopira^®^, Heraeus Kulzer GmbH, Hanau, Germany) was used as local anaesthesia. If necessary, 400 mg of ibuprofen was used as an analgesic.

To access the alveolar crest defect, a mucoperiosteal flap was raised with a mesial or distal-releasing incision. Once the alveolar crest defect was exposed, the implant site was prepared according to the implant manufacturer’s protocol. All implants were placed at bone level (Figure 2a,b) with all implant surfaces fully covered by native bone or the autogenous dentin graft (Figure 2c,d).

The prepared dentin shells were then fixed in front of the alveolar ridge defect with osteosynthesis screws (Microscrews^®^, Stoma, Emmingen-Liptingen, Germany) and the resulting cavity was filled with the prepared particulated dentin (Figure 2c,d). The dentin shell was fixed in such a way that it showed no mobility. This was verified with tweezers. In all cases, the particulated dentin was enough to fill the resulting cavity. If necessary, the mucoperiosteal flap was extended to ensure tension-free wound closure. Supramid^®^ 5-0 (Serag-Wiessner, Naila, Germany) was used as suture material. No bone substitutes or membranes were used during augmentation. After completion of augmentation and implant placement, a CBCT was taken (measurement time T1).

After a 3-month healing period, the implants were exposed by preparation of a small mucosal flap. The osteosynthesis screws were removed in the same session. During implant exposure, the peri-implant bone level was measured with a periodontal probe at four sites (mesial, distal, oral, and buccal) starting from the implant shoulder. Implant stability was checked with a resonance frequency analysis (Ostell Idx, W&H, Buermoos, Austria), and implants with an implant stability quotient (ISQ) of >60 were cleared for prosthetic restoration. Another CBCT was taken (measurement time T2) to check the augmentation success.

### 2.5. Prosthetic Restoration

Four weeks after implant exposure the prosthetic restoration was started. Within the next 4 weeks, all fixed dentures were incorporated so that the whole treatment was completed after 5 months.

### 2.6. Radiographic Evaluation of Bone Gain and Resorption

The changes in the peri-implant bone morphology, especially the buccal bone of the implants, were measured in high-resolution small-volume (50 × 50 mm field of view (FOV)) CBCTs (PaX-Duo3D, Orange Dental, Biberach an der Riß, Germany). The CBCT of T1 was compared with the CBCT of T2. The radiographic measurements included the peri-implant bone level, the thickness of the buccal bone layer on the implant, and the alveolar ridge width. Mesial and/or distal bone loss at the implants was measured at T2 at the mesial and distal edge of the implant shoulder up to the first implant/bone contact (Figure 3a). Only the largest measurement of the two was included for analysis. Buccal bone coverage was examined in a bucco-oral aligned plane (Figure 3b). If no radiologically visible bone coverage was identified, measurements were again taken from the implant shoulder to the first implant/bone contact (Figure 3b). The thickness of the buccal bone layer on the implant was measured at the level of the implant shoulder (L0) and 2 mm (L2) and 4 mm (L4) below the implant shoulder (Figure 4a,b) at T1 and T2. The width of the alveolar crest was measured 2 mm below the implant shoulder (Figure 4c,d) at T1 and T2. All measurements were performed by the same calibrated and blinded examiner using the EZ3D Plus software (Vatech Co. Ltd., Hwaseong-si, Korea).

### 2.7. Outcomes

The primary outcome was biological complications, the secondary outcome was radiographically detectable bone gain surrounding the implants, and the tertiary outcome was osseointegration.

Biological complications were distinguished into severe and non-severe. Severe complications were defined as graft or implant loss during the observation period. Non-severe complications were wound dehiscences and infections with or without suppuration if they did not cause graft or implant loss.

Radiographic changes were evaluated as described in the radiographic evaluation section.

Complete osseointegration was defined as: less than 1 mm bone loss at the four described measurement sites circularly around the implant, ISQ value > 60, implant coverage of radio-opaque structure in the CBCT, and complete buccal bone coverage of the implant in the CBCT (no more than 1 mm loss).

In addition, the following patient-related parameters were extracted from the electronic patient records: age and sex of the patient; periodontal status of the tooth used for augmentation; and implant type, length, diameter, and site.

### 2.8. Statistical Analyses

Data were collected and analysed with IBM SPSS Statistics 22 (SPSS Inc., Chicago, IL, USA).

All evaluations were performed at the patient level, region level, and implant level. In the case of evaluations at the implant level, each implant was evaluated individually. If a patient with two implants had one complication at a single implant, it was scored as one complication. At the implant level, the complication rate was 50% in such a case. In the case of evaluations at the region level, several implants were combined within a sextant, or if they were no more than two tooth regions apart. If there were two implants in one region and one implant had a complication, the complication rate was 100% at the region level. If the two implants were in different regions, the complication rate was 50%. All implants were combined in patient-level evaluations. If a patient had two implants and one of them had a complication, then it was irrelevant whether the two implants were in the same region or not. In both cases, the patient-level complication rate was scored 100%.

Means and standard deviations were calculated at T1 and T2 for the alveolar crest width at L2 and the buccal bone coverage at the implant for L0, L2, and L4. The differences between T1 and T2 at Lx were used to calculate bone resorption.

Fisher’s exact test and cross-tabulations were used for categorical data. Mean values were compared by two-sample *t*-tests.

## 3. Results

A total of 41 patients (22 female, 19 male) were identified who underwent augmentation with TST between January 2019 and March 2020. A total of 58 implants of different types from ASTRA TECH (Astra Tech Implant System, Dentsply Sirona, New York, NY, USA), Nobel Biocare (Nobel Biocare, Kloten, Switzerland), and Conelog (CONELOG^®^, ALTATEC GmbH, Wimsheim, Germany) were used simultaneously. The mean age of the patients was 62.0 (SD 10.2) years. No statistically significant differences in age, gender distribution, initial bone width, implant system, and implant region were found (Table 1). Only the implant diameter was significantly larger in the PCT group.

### 3.1. Severe Complications

During the follow-up period, one augmentation loss due to wound infection occurred in the PCT group (Table 2). In the same patient, a total of two regions were augmented and implanted. Only one augmentation including the implant was lost.

### 3.2. Non-Severe Complications

Three non-severe complications developed during the follow-up period. Two dehiscences were observed in NPCT. One case in the PCT group had an infected inflammation that led to one severe complication with graft and implant loss (Table 2).

No statistically significant differences were found between the treatment groups at the patient, region, or implant level for either the severe or non-severe complications.

### 3.3. Radiographic Evaluation of Bone Gain and Resorption

The CBCT evaluation did not show any bone loss at the mesial or distal implant shoulder, except in the one case where both the graft and implant were lost. In one case in group PCT, a bone loss of 1 mm was observed at the buccal bone coverage, in all other cases except the augmentation loss case, the implants were completely covered by bone.

The mean alveolar ridge width at the patient level was 8.8 mm (SD 1.8) in the PCT group and 8.8 mm (SD 1.4) in the NPCT group (Table 3). At T2, the alveolar ridge widths were 8.5 mm (SD 1.8) and 8.3 mm (SD 1.4), showing resorption of 0.4 mm (SD 0.7) and 0.5 mm (SD 0.7). No statistically significant differences in alveolar ridge widths at T1 and T2 and in resorption were found between the study groups, neither at the patient level, nor region level, nor implant level.

The thickness of buccal bone coverage at the patient level at T1 was 2.7 mm (SD 0.9) at L0, 3.3 mm (SD 1.0) at L2, and 3.6 mm (SD 1.5) at L4 in the PCT group; and 2.7 mm (SD 1.1) at L0, 3.1 mm (1.0) at L2, and 3.2 mm (SD 1.0) at L4 in the NPCT group. At T2, the thicknesses on L0, L2, and L4 were 2.6 mm (1.3), 3.0 mm (1.2), and 3.4 mm (1.6) in the PCT group, and 2.2 mm (1.1), 2.8 mm (SD 0.9), and 2.8 mm (SD 1.1) in the NPCT group respectively (Table 3). This resulted in mean resorptions of 0.1 mm (SD 1.5), 0.3 mm (SD 0.6), and 0.3 mm (0.6) in the PCT group, and 0.5 mm (SD 0.7), 0.4 mm (SD 0.6), and 0.4 mm (SD 0.6) in the NPCT group (Table 4). No statistically significant differences in bone coverage on the implants at T1 and T2 and in resorption were found between the study groups at either patient, region, or implant level. In all cases, buccal bone coverage on the implant surfaces was visible in the CBCT.

The ridge width ratio at patient-level between T1 and T2 was 0.96 (SD 0.08) in the PCT group and 0.94 (SD 0.08) in the NPCT group. For buccal bone coverage thickness, the ratios were 1.13 (SD 1.11), 0.91 (SD 0.17), and 0.92 (SD 0.16) on L0, L2, and L4 in the PCT group; and 0.83 (SD 0.25), 0.90 (SD 0.19), and 0.88 (SD 0.22) in the NPCT group, respectively (Table 5).

### 3.4. Peri-Implant Tissue Probing

The probing depth did not exceed 0.5 mm except in the case where the augmentation was lost.

### 3.5. Implant Stability

The ISQ values measured ranged from 62 to 89 at exposure and were thus >60 in all cases. The mean values were 74 (SD 7.7) in the PCT group and 75 (SD 7.0) in the NPCT group. There were no statistically significant group differences.

### 3.6. Osseointegration

All implants were fully osseointegrated by definition, except for the one that was lost with the graft.

### 3.7. Prosthetic Restoration

All osseointegrated implants could be restored with a fixed restoration.

## 4. Discussion

In the present proof-of-concept study, to our best knowledge we demonstrated, for the first time in humans, that the pre-implantological augmentation with dentin from periodontally compromised teeth leads to similar results compared to augmentation with non-periodontally compromised teeth. Out of a total of 41 augmentations, only one was lost in the PCT group due to severe wound infection, making implant placement impossible. In all other cases, only a few minor complications occurred, which were managed so that implant placement was successful within the 5-month control period. In all cases in which the augmentation healed, sufficient bone width was achieved to allow implant placement. No statistically significant differences were found between the PCT and NPCT groups in terms of severe complications, non-severe complications, bone width achieved, or resorption rates during the observation period.

### 4.1. Severe Complications

The loss of the graft is the most unfavourable event in an implantation that is not possible without bone augmentation. This severe complication leads to corrective surgery or failure of the implantation. In the present study, one dentin graft was lost due to wound infection, which corresponds to a loss rate of approximately 2% regarding all 41 augmentations with dentin. Compared to other studies in which bone block grafting was performed with different techniques, this loss rate is in the lower range of the observed graft loss rate of approximately 0.4–9.1% [14,15,16,17]. In studies investigating different membrane techniques and titanium mesh in various combinations with autogenous bone or bone substitutes, loss rates of approximately 4% were observed [18,19]. In another study of our working group, in which augmentation with the bone block technique fixed at a distance according to Khoury (BBG-D) [15] was compared to the SonicWeld Rx shell technique (SWST) using Poly-D-L-Lactide membranes, a relatively low graft loss rate of approximately 7% was shown in the BBG-D group, while approximately 33% of the grafts were lost in the SWST group [20]. In three other proof-of-concept studies conducted by our research group, in which the tooth-shell technique was fundamentally described and in another compared with the bone-shell technique, as well as in a third study in which the tooth-shell technique with endodontically compromised teeth was compared with non-endodontically compromised teeth, augmentation loss rates of 0–4% were found [1,2,9]. Overall, tooth augmentation with dentin from periodontally compromised teeth thus shows comparable results with other augmentation methods in terms of the graft loss rate, but offers the advantage that, for example, a bone harvesting site with the associated additional morbidity is not required.

### 4.2. Non-Severe Complications

Non-severe complications such as wound dehiscence or inflammation that do not lead to a loss of the implant are unpleasant incidents for the patient in the context of augmentation but do not lead to a loss of the implant. Wound dehiscence has been described in association with various augmentation techniques, especially when membranes were used, with rates of up to 50% [21,22,23,24,25]. For bone block grafting, dehiscence rates are generally below 7% [15,16,17]. The dehiscence rates found in the other tooth augmentation studies of our working group ranged in a comparable area below 7% [1,2,9]. In respective animal studies, dehiscence rates of 20% and 29% [7,8] were described, which were considerably higher than the usually described dehiscence rates in bone grafting in humans and considerably higher than the overall dehiscence rate of around 5% at the patient level in the present study.

Reports concerning inflammation in grafting procedures are rather rare. Two studies report 5.8% and 3.2% of infections at the augmentation site in bone grafting and in augmentations with membranes and autologous bone mixed with a bone substitute material [16,18]. In the above-mentioned proprietary BBG-D vs. SWST study, approximately 7% of infections were observed across all study groups [20]. The overall rate of 2% in the present study is thus very low.

Concerning non-severe complications, we can conclude that augmentation with the dentin of periodontally damaged teeth has a similarly low, if not lower, complication rate than other grafting methods.

### 4.3. Radiographic Evaluation of Bone Gain

The aim of the augmentation procedure in lateral alveolar bone deficiency is to gain bone width sufficient for subsequent implant placement. Accordingly, many publications are primarily concerned with bone gain achieved by augmentation. It was sufficiently proven that different augmentation techniques lead to sufficient bone gain to make implant placement possible [26]. However, it is also known that resorption rates can be high, up to 18%, when augmenting with bone blocks [27,28]. In the present study, a ridge width of 8.8 mm was measured immediately after augmentation in both study groups. In the BBG-D vs SWST study, bone widths of 8.9 mm and 9.0 mm, respectively, were measured [20]. In the other tooth augmentation studies of our study group, bone widths of 8.7 mm to 9.5 mm were found [1,9]. The bone widths immediately after augmentation differed slightly by up to 0.7 mm, which is not clinically relevant given the total bone width achieved. More interesting, however, are the resorption rates. The resorption of the total ridge width in the BBG-D vs. SWST study after 4 months was 1.1 mm in the BBG-D group and even 2.2 mm in the SWST group [20], while in all of our tooth augmentation studies, including the present one, the resorption of the alveolar ridge width after 3 months was not more than 0.5 mm [1,9]. This is a considerable difference of more than one-half the resorption in a comparable period. The buccal bone lamellae were similarly stable at all investigated levels on the implant in the present study as well as in our other tooth augmentation studies [1,9].

It can thus be concluded that the augmentation of alveolar ridge deficits with the Tooth-shell technique leads to stable augmentations with a lower tendency to resorption than other augmentation techniques. 

### 4.4. Osseointegration and Implant Stability

Verification of osseointegration by the histological investigation was not possible in the present study. Accordingly, a resonance frequency analysis was performed to determine the implant stability quotient (ISQ). In general, it can be assumed that from ISQ values of 60 upwards, an implant is integrated into the bone and can be loaded prosthetically [29]. In the present study, ISQ values of 74 and 75 were achieved in the PCT and NPCT groups. This is comparable to the other tooth augmentation studies of our working group [1,2,9]. In addition, a probing depth from the implant shoulder to the bone not greater than 0.5 mm was recorded when the implants were uncovered. Accordingly, we concluded that all implants were fully osseointegrated at the time of exposure.

### 4.5. Infection in Periodontally Compromised Teeth

This study aimed to investigate whether periodontally compromised teeth are also suitable for pre-implantological augmentation. It should be borne in mind that, in contrast to non-compromised teeth, an infection of the dentin can always be present. As already shown in another study of our working group, endodontically treated teeth are similarly suitable for augmentation purposes like non-endodontically treated teeth. While infection of dentin in endodontically treated teeth results from the pulp chamber and root canal inside the tooth, potential infection in periodontally compromised teeth can be expected from the root surface. While endodontic studies have shown that associated infection allows penetration of bacteria into the dentinal tubules from within the tooth [30], similar penetration of bacteria into the dentin from the root surface has been observed in periodontally compromised teeth [11]. Although the endodontic and periodontal infection profiles differ, the same pathogens can generally be found in different frequencies [31]. In the animal studies already mentioned above, relatively high loss rates of augmentations were observed with endodontically treated and periodontally compromised teeth [7,8]. However, no protocol for decontamination of the potentially infected dentin was described in these studies. In contrast, our studies were conducted with adherence to a disinfection protocol for the tooth components used. In the present study, the same dentin preparation technique was used as in the study with endodontically treated teeth. Very similar study results were obtained. Accordingly, we assumed that the disinfection protocol used achieves a depth of activity that results in sufficient decontamination of potentially infected dentin so that healing of the augmentation is possible without major complications.

### 4.6. Limitations

The present study is a proof-of-concept study. Thus, there are certain limitations. On the one hand, the study could not be randomized, and on the other hand, it was a retrospective convenience sample. However, randomization is difficult to imagine in this type of study since a periodontally compromised tooth and a non-compromised tooth are usually not simultaneously available for augmentation in the same patient for randomization. However, since the documentation of the treatment cases has a prospective character and includes all relevant parameters that are necessary for the evaluation of the augmentation and implantation success, we believe that the quality of a prospective study is almost achieved. Another limitation of this study was the relatively short follow-up period of 5 months. Longer follow-up periods are necessary for a better assessment of resorption.

## 5. Conclusions

Within the limitations of this study, augmentation of alveolar ridge defects with dentin from periodontally compromised teeth showed similar results to augmentation with non-periodontally compromised teeth. Augmentation with dentin from periodontally compromised teeth is thus a clinically useful addition to augmentation methods that avoid a bone-harvesting site and thus reduce patient-related morbidity associated with augmentation procedures.

## Figures and Tables

**Figure 1 ijerph-19-04560-f001:**
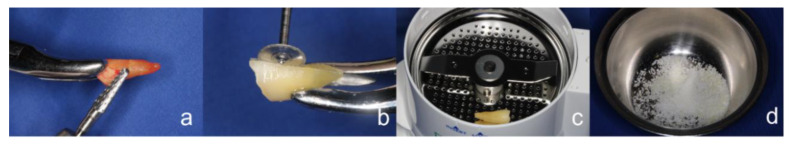
Dentin preparation. (**a**): A carbide bur was used to remove periodontal tissue and debris from the tooth root. In addition, restorations were eliminated. (**b**): A dentin shell was separated from the tooth root with a rotating diamond disc. (**c**): The remaining dentin was particulated in a sterile disposable mill (Smart Dentin Grinder). (**d**): The particulated dentin was kept in a sterile container.

**Figure 2 ijerph-19-04560-f002:**
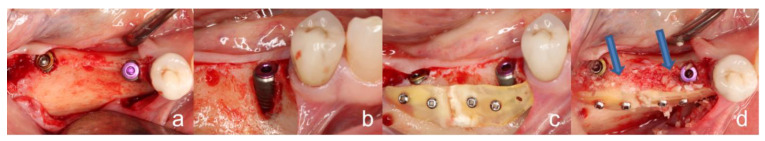
Surgical procedure of the tooth shell technique. (**a**,**b**): Occlusal and lateral view: inserted implants in regions 45 and 47. The vestibular implant surface of region 45 was not covered with bone. (**c**): Vestibular of the two implants, two dentin shells were fixed with four osteosynthesis screws. (**d**): The created cavity between the dentin shells and the residual bone or implants were filled with particulate dentin (blue arrows).

**Figure 3 ijerph-19-04560-f003:**
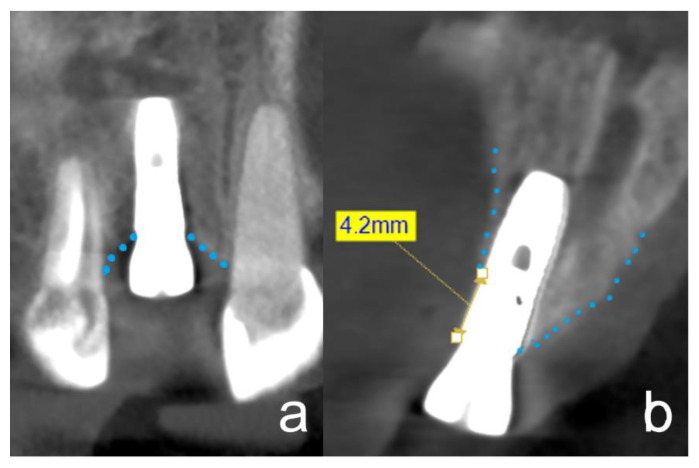
CBCT 3 months after implant insertion, at the time of implant exposure. The dotted lines outline the alveolar ridge. The images show distinct cases to demonstrate the respective measurement methods. (**a**): The frontal plane in the CBCT is equivalent to the two-dimensional X-ray image. At the time of implant exposure, there was no horizontal bone loss in either the mesial or distal area. (**b**): This figure is an example to show the measurement technique of resorption of the buccal lamella, if it was present at all. The measurements were taken from the implant shoulder to the first contact between the implant and the hard tissue. In this figure, an exemplary resorption of the buccal lamella of 4.2 mm can be seen.

**Figure 4 ijerph-19-04560-f004:**
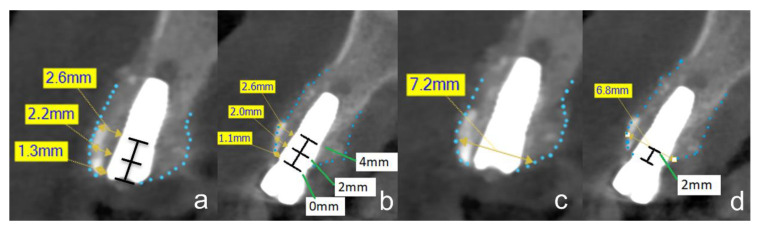
CBCT at time of implant insertion (**a**,**c**) and three months after implant insertion, at the time of implant exposure (**b**,**d**). The dotted lines outline the alveolar ridge. The images show the same case with the respective measurement methods. (**a**,**b**): The assessment and measurement of the buccal lamella are shown in this figure. The tooth shell is visible in the vestibular area. The thickness of the buccal lamella was measured at three different levels (L0 = 0 mm, L2 = 2 mm, and L4 = 4 mm). (**c**,**d**): The bucco-oral alveolar ridge width was also measured in the sagittal plane. The measurement was carried out at level L2. In this case, the width was 6.8 mm.

**Table 1 ijerph-19-04560-t001:** Baseline data at the time of augmentation.

		Study Group		Sign
Baseline Data of Participants	Total	PCT	NPCT	*p*-Value
**Age (years)**				
**Mean (SD)**	62.0 (10.2)	61.9 (7.5)	62.0 (12.5)	n.s.
Range	28–80	49–76	28–80	
**Gender (male)**				
*n* (%)	19 of 41 (46)	10 of 19 (53)	9 of 22 (41)	n.s.
**Mean initial bone width (mm)** Mean (SD)	3.5 (1.6)	3.5 (1.8)	3.5 (1.4)	n.s.
**Implantat system** Astra Tech *n*	34	15	19	n.s.
Conelog *n*	4	3	1	n.s.
Nobel Biocare *n*	20	11	9	n.s.
**Implant diameter (mm)** Range Mean (SD)	3.5–5 4.1 (0.38)	3.5–4.8 4.2 (0.36)	3.6–5 4.0 (0.39)	n.s. 0.046
**Implant region** Anterior region (13–23, 33–43) *n* Posterior region (14–17, 24–27, 34–37, 44–47) *n*	19 39	9 20	10 19	n.s. n.s.

PCT = Periodontally compromised teeth, NPCT = Non-periodontally compromised teeth, n.s. = not significant.

**Table 2 ijerph-19-04560-t002:** Clinical complications on a patient, region, and implant level.

		Study Group		Fisher’s Exact Test (2-Sided)
Clinical Complication	Total	PCT	NPCT	*p*-Value
**Total severe complications**				
*n* (%) on PL	1 of 41 (2)	1 of 19 (5)	0 of 22 (0)	0.463
*n* (%) on RL	1 of 51 (2)	1 of 26 (4)	0 of 25 (0)	1.000
*n* (%) on IL	1 of 58 (2)	1 of 29 (3)	0 of 29 (0)	1.000
**Wound dehiscence**				
*n* (%) on PL	2 of 41 (5)	0 of 19 (0)	2 of 22 (9)	0.490
*n* (%) on RL	2 of 51 (4)	0 of 26 (0)	2 of 25 (8)	0.235
*n* (%) on IL	2 of 58 (3)	0 of 29 (0)	2 of 29 (7)	0.491
**Inflammation (pus)**				
*n* (%) on PL	1 of 41 (2)	1 of 19 (5)	0 of 22 (0)	0.463
*n* (%) on RL	1 of 51 (2)	1 of 26 (4)	0 of 25 (0)	1.000
*n* (%) on IL	1 of 58 (2)	1 of 29 (3)	0 of 29 (3)	1.000
**Total complications at all**				
*n* (%) on PL	3 of 41 (7)	1 of 19 (5)	2 of 22 (9)	1.000
*n* (%) on RL	3 of 51 (6)	2 of 26 (8)	1 of 25 (4)	0.610
*n* (%) on IL	3 of 58 (5)	1 of 29 (3)	2 of 29 (7)	1.000

PCT = Periodontally compromised teeth, NPCT = Non-periodontally compromised teeth, PL = patient level, RL = region level, IL = implant level.

**Table 3 ijerph-19-04560-t003:** Mean values of the CBCT evaluations at time points T1 and T2.

		Study Group		Two-Sample *t* Test
Time of Measurement	Mean	PCT	NPCT	(*p*-Value)
**T1**				
**Mean bucco-palatal alveolar ridge width (mm)**				
PL, *n* = 41 (SD)	8.8 (1.6)	8.8 (1.8)	8.8 (1.4)	0.918
RL, *n* = 51 (SD)	8.7 (1.6)	8.7 (1.8)	8.7 (1.4)	0.920
IL, *n* = 58 (SD)	8.7 (1.6)	8.7 (1.8)	8.7 (1.4)	0.962
**Mean buccal lamella width L0 (mm)**				
PL, *n* = 41 (SD)	2.7 (1.0)	2.7 (0.9)	2.7 (1.1)	0.928
RL, *n* = 51 (SD)	2.6 (1.0)	2.6 (0.9)	2.6 (1.1)	0.986
IL, *n* = 58 (SD)	2.6 (1.0)	2.7 (0.9)	2.6 (1.1)	0.812
**Mean buccal lamella width L2 (mm)**				
PL, *n* = 41 (SD)	3.2 (1.0)	3.3 (1.0)	3.1 (1.0)	0.718
RL, *n* = 51 (SD)	3.1 (1.0)	3.2 (1.0)	3.0 (1.0)	0.645
IL, *n* = 58 (SD)	3.1 (1.0)	3.2 (1.0)	3.0 (1.0)	0.471
**Mean buccal lamella width L4 (mm)**				
PL, *n* = 41 (SD)	3.4 (1.3)	3.6 (1.6)	3.2 (1.0)	0.393
RL, *n* = 51 (SD)	3.4 (1.3)	3.5 (1.5)	3.2 (1.0)	0.340
IL, *n* = 58 (SD)	3.4 (1.3)	3.6 (1.5)	3.1 (0.9)	0.184
**T2**				
**Mean bucco-palatal alveolar ridge width (mm)**				
PL, *n* = 41 (SD)	8.4 (1.5)	8.5 (1.8)	8.3 (1.4)	0.634
RL, *n* = 50 (SD)	8.3 (1.5)	8.3 (1.8)	8.2 (1.3)	0.752
IL, *n* = 57 (SD)	8.3 (1.6)	8.3 (1.8)	8.2 (1.4)	0.671
**Mean buccal lamella width L0 (mm)**				
PL, *n* = 41 (SD)	2.4 (1.2)	2.6 (1.3)	2.2 (1.1)	0.259
RL, *n* = 50 (SD)	2.3 (1.2)	2.4 (1.3)	2.2 (1.1)	0.577
IL, *n* = 57 (SD)	2.3 (1.1)	2.4 (1.2)	2.2 (1.0)	0.408
**Mean buccal lamella width L2 (mm)**				
PL, *n* = 41 (SD)	2.9 (1.0)	3.0 (1.2)	2.8 (0.9)	0.496
RL, *n* = 50 (SD)	2.8 (1.0)	2.9 (1.1)	2.7 (0.9)	0.592
IL, *n* = 57 (SD)	2.8 (1.0)	2.9 (1.1)	2.7 (0.9)	0.433
**Mean buccal lamella width L4 (mm)**				
PL, *n* = 41 (SD)	3.1 (1.3)	3.4 (1.6)	2.8 (1.1)	0.209
RL, *n* = 50 (SD)	3.1 (1.3)	3.4 (1.5)	2.8 (1.0)	0.135
IL, *n* = 57 (SD)	3.1 (1.3)	3.4 (1.5)	2.8 (1.0)	0.083

*n* = number, SD = standard deviation, PCT = Periodontally compromised teeth, NPCT = Non-periodontally compromised teeth, PL = patient level, RL = region level, IL = implant level.

**Table 4 ijerph-19-04560-t004:** Mean values of resorption from T1 to T2.

		Study Group		Two-Sample *t* Test
Mean Resorption in mm	Mean	PCT	NPCT	(*p*-Value)
**bucco-oral alveolar ridge**				
PL, *n* = 41 (SD)	0.4 (0.7)	0.4 (0.7)	0.5 (0.7)	0.461
RL, *n* = 50 (SD)	0.4 (0.8)	0.4 (0.8)	0.5 (0.7)	0.436
IL, *n* = 57 (SD)	0.4 (0.7)	0.4 (0.8)	0.5 (0.7)	0.472
**L0**				
PL, *n* = 41 (SD)	0.3 (1.1)	0.1 (1.5)	0.5 (0.7)	0.270
RL, *n* = 50 (SD)	0.3 (1.1)	0.3 (1.4)	0.4 (0.7)	0.550
IL, *n* = 57 (SD)	0.3 (1.0)	0.2 (1.3)	0.4 (0.7)	0.487
**L2**				
PL, *n* = 41 (SD)	0.3 (0.6)	0.3 (0.5)	0.4 (0.6)	0.596
RL, *n* = 50 (SD)	0.3 (0.6)	0.3 (0.5)	0.3 (0.6)	0.970
IL, *n* = 57 (SD)	0.3 (0.5)	0.3 (0.5)	0.3 (0.6)	0.999
**L4**				
PL, *n* = 41 (SD)	0.3 (0.6)	0.3 (0.5)	0.4 (0.6)	0.517
RL, *n* = 50 (SD)	0.3 (0.6)	0.3 (0.5)	0.4 (0.6)	0.516
IL, *n* = 57 (SD)	0.3 (0.5)	0.3 (0.5)	0.3 (0.6)	0.738

*n* = number, SD = standard deviation, PCT = Periodontally compromised teeth, NPCT = Non-periodontally compromised teeth, PL = patient level, RL = region level, IL = implant level.

**Table 5 ijerph-19-04560-t005:** Ratio of residual bone width of bucco-oral alveolar ridge and buccal lamella from T1 to T2.

		Study Group		Two-Sample *t*-Test
Ratio from T1 to T2	Mean	PCT	NPCT	(*p*-Value)
**bucco-oral alveolar ridge**				
PL, *n* = 41 (SD)	0.95 (0.08)	0.96 (0.08)	0.94 (0.08)	0.471
RL, *n* = 50 (SD)	0.95 (0.08)	0.96 (0.09)	0.94 (0.08)	0.438
IL, *n* = 57 (SD)	0.95 (0.08)	0.96 (0.09)	0.94 (0.08)	0.427
**L0**				
PL, *n* = 41 (SD)	0.96 (0.78)	1.13 (1.11)	0.83 (0.25)	0.231
RL, *n* = 50 (SD)	0.93 (0.71)	1.01 (0.98)	0.85 (0.25)	0.406
IL, *n* = 57 (SD)	0.93 (0.68)	1.02 (0.93)	0.84 (0.25)	0.330
**L2**				
PL, *n* = 41 (SD)	0.90 (0.18)	0.91 (0.17)	0.90 (0.19)	0.957
RL, *n* = 50 (SD)	0.91 (0.19)	0.89 (0.18)	0.92 (0.20)	0.574
IL, *n* = 57 (SD)	0.91 (0.18)	0.90 (0.17)	0.92 (0.19)	0.578
**L4**				
PL, *n* = 41 (SD)	0.90 (0.19)	0.92 (0.16)	0.88 (0.22)	0.489
RL, *n* = 50 (SD)	0.91 (0.19)	0.93 (0.18)	0.89 (0.21)	0.432
IL, *n* = 57 (SD)	0.91 (0.19)	0.93 (0.17)	0.90 (0.20)	0.607

*n* = number, SD = standard deviation, PCT = Periodontally compromised teeth, NPCT = Non-periodontally compromised teeth, PL = patient level, RL = region level, IL = implant level.

## Data Availability

Data is contained within the article.

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
