# Peer review of "Lateral Alveolar Ridge Augmentation with Autologous Dentin of Periodontally Compromised Teeth: A Retrospective Study"

_ijerph, 2022, doi:10.3390/ijerph19084560_

Round 1
Reviewer 1 Report
I have a number of concerns which should be adequately addressed:
- In fact you performed a proof of concept study and not a retrospective controlled study. Otherwise, make clear how it was controlled?
- Why did you not perform a prospective study, as you performed a controlled study?
- In lines 102-103 you wrote: For this study, all cases with a bone defect of at least 4 mm were included, so that a hard tissue gain of 4 mm or more in width had to be achieved by augmentation. Does this mean that you only included severe cases? What width of the alveolar process was accepted?
- How many implants were fully covered with native bone, how many needed extensive coverage with dentin particles.
- Were the screws removed after 3 months?
- Fig. 3: there seems no bone around the total implant surface. Why not?
- Add also to table 1 the type of implants placed, the region of placement and the seize of the defects. Otherwise, any comparison can be biased or a conclusion cannot be drawn.
- Table 2, please check the p values. p=1.000 is strange when there are differences between the groups.
- Add also T0 to table 3. That tell something about the defects to be augmented and the comparison of the defects.
- Randomization would be possible between a group of periodontal diseased and healthy patients and evaluating by a researcher not involved in the implant placement and/or prosthodontics. He/she could be blinded.
Author Response
Reviewer 1
I have a number of concerns which should be adequately addressed:
- In fact you performed a proof of concept study and not a retrospective controlled study. Otherwise, make clear how it was controlled?
> There are two groups in the present study, that were retrospectively analyzed. One group in which periodontally damaged/compromised teeth were used for augmentation and a second group in which non-periodontally damaged teeth were used for augmentation. The aim of this study was to investigate whether the use of periodontally damaged teeth for preimplantologial augmentation leads to comparable results as the use of periodontally healthy teeth. Therefore, the second group acted as a control group. Since we have evaluated existing data according to the study objective mentioned above, this is a retrospective study.
- Why did you not perform a prospective study, as you performed a controlled study?
> As previously described, we examined existing data of treatments in the past for the study objective we stated. Therefore, this is a retrospective and not a prospective study.
- In lines 102-103 you wrote: For this study, all cases with a bone defect of at least 4 mm were included, so that a hard tissue gain of 4 mm or more in width had to be achieved by augmentation. Does this mean that you only included severe cases? What width of the alveolar process was accepted?
> It is correct that only cases with a preimplantological bone deficit of at least 4 mm were included in the study. The initial bone width was varying. It was important to us that the augmentation should have a minimum size of 4 mm. Thus, only severe atrophic cases were evaluated. The minimum planned implant diameter was 3.5mm. In this case, the desired bone width should be 6.5mm (3.5mm + 1.5mm buccal + 1.5 mm oral). With a bone deficit of at least 4mm, the initial bone width was in such a case a maximum of 2.5 mm.
- How many implants were fully covered with native bone, how many needed extensive coverage with dentin particles.
> Not a single implant included in the study was completely covered with native bone. All implants included in the study had a bone deficit of at least 4 mm. We mentioned this at M&M.
- Were the screws removed after 3 months?
>The screws were removed after 3 months. The following has been added to the text: “The osteosynthesis screws were removed in the same session.”
- Fig. 3: there seems no bone around the total implant surface. Why not?
>Figure 3b was just an example to better illustrate the measurement-technique for a potential case with buccal bone loss. The following has been added to the text: “This figure is an example to show the measurement of resorption of the buccal lamella.“
- Add also to table 1 the type of implants placed, the region of placement and the seize of the defects. Otherwise, any comparison can be biased or a conclusion cannot be drawn.
>We added the initial bone width, implant system, implant diameter and implant region (anterior and lateral) to Table 1
- Table 2, please check the p values. p=1.000 is strange when there are differences between the groups.
> The p-values ​​between the groups were not different. However, it was evaluated at three different levels (patient level, region level and implant level). Since the number was of course different for the different levels, there were different p-values. We checked the p-values. These are correct.
- Add also T0 to table 3. That tell something about the defects to be augmented and the comparison of the defects.
>We have added the initial bone width in Table 1. Therefore these values ​​are not repeated in Table 3.
- Randomization would be possible between a group of periodontal diseased and healthy patients and evaluating by a researcher not involved in the implant placement and/or prosthodontics. He/she could be blinded.
> As far as we know, randomization between a group of periodontally diseased and healthy patients is not possible, since the conditions are different and cannot be changed. Either you have periodontitis or you don't. It was also not possible to blind the surgeon because the surgeon removed the tooth and performed the augmentation. Thus, the surgeon knew the patient. Moreover, we did not evaluate patients with periodontitis. We evaluated the use of either periodontally compromised teeth or non-compromised teeth. Even in this case randomization is not possible, because a patient has either a compromised or a non-compromised tooth. A simple blinding during the evaluation is of course possible. The evaluator was blinded in this study. We have added this information to M&M.
Reviewer 2 Report
The authors report the clinical case of augmentation using tooth shell technique for implant treatment. The technique is unique and I guess that this report will be of interest to its readership. However, I guess also that the following issues should be revised or mentioned for publish;
In material and methods
- All tooth used in this study had no history of endodontic treatment?
- In line 125, the total chemical cleaning time using sodium hydroxide and ethanol was 10 minutes? Or each chemical cleaning of those solution was 10 minutes?
- The screws were removed or remained in formed tissue after three months healing periods?
- In line 165, “the implanted were exposed” means that gingival flap was prepared and bone healing was directly observed? or for mounting abutment?
- The addition of the images of CBCT of T1 and T2 make easy to show the effect of this treatment for comparison.
- The volume of prepared dentin particulates was enough for filling bone defects at the level of cover fully an implant surface in all case?
- The area of the exposed implant surface on native bone was regulated or depend on the case?
- The letters position should be regulated in table3-5.
- In evaluation, authors should be more clear what is “at the patient level, implant region level and implant level.” In particular, it is difficult to image what did authors evaluate at the patient level on the evaluation of resorption or ratio of residual bone width from the CBCT images.
Discussion
- I guess that 5 months is too short for the observation period of the resorption of bone healing. Authors should mention about the healing period like why authors set 5 months.
Author Response
Reviewer 2
The authors report the clinical case of augmentation using tooth shell technique for implant treatment. The technique is unique and I guess that this report will be of interest to its readership. However, I guess also that the following issues should be revised or mentioned for publish;
In material and methods
- All tooth used in this study had no history of endodontic treatment?
> There were no teeth with a root canal treatment. Otherwise it would have been mentioned in the inclusion criteria.
- In line 125, the total chemical cleaning time using sodium hydroxide and ethanol was 10 minutes? Or each chemical cleaning of those solution was 10 minutes?
> The first step took 10 minutes, the second and third step 3 minutes each. The total time was 16 minutes. We added this information to the text.
- The screws were removed or remained in formed tissue after three months healing periods?
>>The screws were removed after 3 months. The following has been added to the text: “The osteosynthesis screws were removed in the same session.”
- In line 165, “the implanted were exposed” means that gingival flap was prepared and bone healing was directly observed? or for mounting abutment?
> A small mucosal flap was formed at implant exposure. The osseointegration was checked in the same session.
- The addition of the images of CBCT of T1 and T2 make easy to show the effect of this treatment for comparison.
>We have included a comparative DVT section in the figure.
- The volume of prepared dentin particulates was enough for filling bone defects at the level of cover fully an implant surface in all case?
> In all cases the particulated dentin was sufficient for the filling of the bone defect.
- The area of the exposed implant surface on native bone was regulated or depend on the case?
> For this study, all cases with a bone defect of at least 4 mm were included, so that a hard tissue gain of 4 mm or more in width had to be achieved by augmentation. The initial bone width was varying. It was important to us that the augmentation should have a minimum size of 4 mm. Thus, only severe atrophic cases were evaluated. The minimum planned implant diameter was 3.5mm. In this case, the desired bone width should be 6.5mm (3.5mm + 1.5mm buccal + 1.5 mm oral). With a bone deficit of at least 4mm, the initial bone width was in such a case a maximum of 2.5 mm.
- The letters position should be regulated in table3-5.
>We changed that.
- In evaluation, authors should be more clear what is “at the patient level, implant region level and implant level.” In particular, it is difficult to image what did authors evaluate at the patient level on the evaluation of resorption or ratio of residual bone width from the CBCT images.
> We have included the following explanation: “In the case of evaluations at the implant level, each implant was evaluated individually. If a patient with two implants, had one complication at a single implant, it was scored as one complication. At the implant level, the complication rate was 50%. In the case of evaluations at region level, several implants were combined within a sextant, or if they were no more than two tooth regions apart. If there were two implants in one region and one implant had a complication, the complication rate was 100%. If the two implants were in different regions, the complication rate was 50%. All implants were combined in patient-level evaluations. If a patient had two implants and one of them had a complication, then it was irrelevant whether the two implants were in the same region or not. In both cases, the patient-level complication rate was 100%.”
Discussion
- I guess that 5 months is too short for the observation period of the resorption of bone healing. Authors should mention about the healing period like why authors set 5 months.
> It is correct that 5 months is short for assessing resorption. Therefore, further studies with a longer follow-up period are necessary. This is one of the limitations of the study. In this proof-of-concept study, we only wanted to assess whether periodontally compromised teeth lead to the same results when using TST as periodontally healthy teeth regarding the possibility to place implants The assessment of the buccal lamella during the observation period was important for us when assessing the osseointegration.
We added the following to the limitations: “Another limitation of this study was the relatively short follow-up period of 5 months. Longer follow-up periods are necessary for a better assessment of resorption.”
Round 2
Reviewer 1 Report
I would not name it a controlled retrospective study as not two groups of patients were studied which were consecutive were selective and assigned to a study group, two groups of patients who had either an periodontal diseased tooth or not were created. There is a hazard of an incorporation bias. So, omit controlled.
Fig. 3 B shows a tooth on one side not covered with bone or at least not a sufficient layer of bone. Was this often the case and how do these implants perform on the long term? Provide otherwise a better example covered with sufficient bone.
In table 1, is the significance level correct at the implant diameter? If so, why was the implant diameter of implant to be placed in NPCT sites lower?
I do not expect a p value of 1.000 when there are differences between groups, it can be close to 1.000, but not 1.000.
Author Response
XCV
Reviewer 1
I would not name it a controlled retrospective study as not two groups of patients were studied which were consecutive were selective and assigned to a study group, two groups of patients who had either a periodontal diseased tooth or not were created. There is a hazard of an incorporation bias. So, omit controlled.
> We removed controlled from the manuscript
Fig. 3 B shows a tooth on one side not covered with bone or at least not a sufficient layer of bone. Was this often the case and how do these implants perform on the long term? Provide otherwise a better example covered with sufficient bone.
> Figure 3b shows an implant. The figure belongs to the "Materials and Methods" section. It only shows the measurement methods used and is explicitly not part of the results section! In the results section we report in the second sentence under "Radiographic evaluation of bone gain and resorption" that 1mm of buccal bone loss was visible on one implant (according to the measurement method demonstrated in Fig. 3 b). This is therefore a rare case. The phenomenon has been described by our group as well as others. The results are stable over time. For a first insight, please read No. 20 of the reference list of this manuscript to start your evidence search.
In table 1, is the significance level correct at the implant diameter? If so, why was the implant diameter of implant to be placed in NPCT sites lower?
> We cannot identify any obvious reason for this phenomenon, as the mean values of the achieved bone widths in PCT and NPCT do not differ. Only the standard deviation in the PCT group is slightly larger. I.e. individual bone widths achieved could be larger in PCT than in NPCT. It was therefore possible to use some implants with a larger diameter in the PCT group. This leads to a statistically significant difference in this case. This is a good example of the fact that not every statistically significant difference has a meaningful clinical relevance. In this case it has no relevance and is therefore irrelevant for the report.
I do not expect a p value of 1.000 when there are differences between groups, it can be close to 1.000, but not 1.000.
> Thank you for your comment. We acknowledge that actually only exactly equal groups lead to a p‑value of 1. Nevertheless, the calculations with leading statistical software result in the value 1. We therefore report our values accordingly in order to avoid irritating other readers who want to follow the calculations in standard statistical software. We therefore strongly recommend against rounding off the p-values to 0.9999 in order to avoid irritation.
We calculated the significance values with both IBM SPSS 22 and BlueSky Statistics v10. Here is the results table from BlueSky for “Total severe complications” “n (%) on RL” from Table 2:
Please check the Word dokument
This is the same result as with SPSS. Moreover, other calculators give the same result.